# Microwave-Assisted One-Pot Lipid Extraction and Glycolipid Production from Oleaginous Yeast *Saitozyma podzolica* in Sugar Alcohol-Based Media

**DOI:** 10.3390/molecules26020470

**Published:** 2021-01-18

**Authors:** André Delavault, Katarina Ochs, Olga Gorte, Christoph Syldatk, Erwann Durand, Katrin Ochsenreither

**Affiliations:** 1Technical Biology, Institute of Process Engineering in Life Sciences II, Karlsruhe Institute of Technology, 76131 Karlsruhe, Germany; katarina.ochs@kit.edu (K.O.); olga.gorte@kit.edu (O.G.); christoph.syldatk@kit.edu (C.S.); katrin.ochsenreither@kit.edu (K.O.); 2CIRAD, UMR QualiSud, F-34398 Montpellier, France; erwann.durand@cirad.fr; 3QualiSud, Univ Montpellier, CIRAD, Institut Agro, Univ Avignon, Univ Réunion, 34000 Montpellier, France

**Keywords:** glycolipid, single cell oil, lipase, microwave, one-pot process, deep eutectic solvents

## Abstract

Glycolipids are non-ionic surfactants occurring in numerous products of daily life. Due to their surface-activity, emulsifying properties, and foaming abilities, they can be applied in food, cosmetics, and pharmaceuticals. Enzymatic synthesis of glycolipids based on carbohydrates and free fatty acids or esters is often catalyzed using certain acyltransferases in reaction media of low water activity, e.g., organic solvents or notably Deep Eutectic Systems (DESs). Existing reports describing integrated processes for glycolipid production from renewables use many reaction steps, therefore this study aims at simplifying the procedure. By using microwave dielectric heating, DESs preparation was first accelerated considerably. A comparative study revealed a preparation time on average 16-fold faster than the conventional heating method in an incubator. Furthermore, lipids from robust oleaginous yeast biomass were successfully extracted up to 70% without using the pre-treatment method for cell disruption, limiting logically the energy input necessary for such process. Acidified DESs consisting of either xylitol or sorbitol and choline chloride mediated the one-pot process, allowing subsequent conversion of the lipids into mono-acylated palmitate, oleate, linoleate, and stearate sugar alcohol esters. Thus, we show strong evidence that addition of immobilized *Candida antarctica* Lipase B (Novozym 435^®^), in acidified DES mixture, enables a simplified and fast glycolipid synthesis using directly oleaginous yeast biomass.

## 1. Introduction

Surfactants are chemicals that are in an ever growing and worldwide demand in various fields of application like pharmaceutics, cosmetics, cleaning agents, or even the petroleum industry [1]. It is also reasonable to think that the recent Covid-19 crisis might boost this demand because of the use of such compounds in washing and more generally in hygiene products [2]. Such context makes it crucial to design processes in agreement with green chemistry principles that aim to synthetize amphiphilic molecules that are not only bio-sourced, but also biodegradable, using innovative and low energy input methods [3]. Glycolipids are in this regard an interesting class of surfactants since they are obtained via chemical or bio-chemical combination of a sugar, or sugar alcohol moiety, and a hydrophobic tail which can be a fatty acid, or a fatty alcohol. They are viable allies for greener surfactant production as the chemical functions that link the polar head and apolar tail are inherently biodegradable [4,5]. Indeed, in the chemical function bonding, those two components can be an ester when a lipase is supplied with a fatty acid playing the substrate role. Certain lipases, or acyltransferases, present the advantage to be useable in mild conditions and allow sugar ester synthesis when placed in water free conditions. We present here a potentially low energy input and simple process combined with a fast microwave-assisted preparation of low water content Deep Eutectic Systems (DESs) that uses constant and accurate control of the temperature coupled with an efficient diffusion of heat. Microwave-assisted chemistry is an environmentally friendly approach that can overcome limitations such as uneven heating, poor mixing, thermal stability of reagents, and high viscosity [6]. Such technology can be considered as a powerful, versatile, and flexible tool for chemical reactions, biomass treatment, and biocatalysis, which match with deep eutectic systems [7].

DESs were first proposed by Abbott et al. as alternative solvents to ionic liquids (ILs) [8]. At a certain molar ratio of the components, there is a significant lowering of the melting point, which results in the formation of a liquid at room temperature [9]. Our one-pot process used an acidified DES made of choline chloride and sugar alcohols, such as xylitol and sorbitol, to extract lipids from oleaginous biomass (*S. podzolica* (*Saitozyma podzolica*) DSM 27192). Subsequently, sugar alcohol esters were produced through lipase-catalyzed reaction. Palmitate, stearate, oleate, and linoleate monoesters were identified by spectrometric and spectroscopic methods. The overall strategy of our method is shown in the flowchart of Figure 1. A similar process was demonstrated by Siebenhaller et al., in which sugar esters were synthesized using a DES based on renewables. However, such integrated production required many steps like any other “Stop-and-go” process [10]. Parallelly, lipid extraction of oleaginous yeast cells is mostly carried out by a combination of cell disruption and extraction processes [11]. Mechanical, physical, chemical, and enzymatic methods have been used for cell disruption [12]. The most common methods for extraction are solvent system-based using chloroform-methanol [13] or ethanol-hexane [14]. It appeared therefore logical to look alternatively for greener and more innovative systems.

Previous work from Gorte et al. demonstrated, in this aim, the use of several methods for the pre-treatment of the biomass produced by *S. podzolica*, including multi-step use of mechanical techniques such as ball milling, ultra-sonication, and high–pressure homogenization [11]. In the present work, we shortened the biomass treatment process through a single step approach, after freeze-drying the biomass, removing therefore the cell disruption step. We introduced the media, the biomass, and the enzyme, all together in one reactor vessel assisted by microwave irradiation. Thus, we reduced the number of steps, and allowed both the lipid extraction and the production of sugar alcohol esters without stopping the reaction or changing the apparatus. The extraction efficiency was compared to a standard Folch extraction, and the amounts, along with the profile of extracted lipids, were investigated.

It was originally expected that enzymatic or auto-catalytic acylation reaction between sugar alcohols composing the DESs and fatty acids of the biomass would have happened without prior modification of the components. Acidified DES (AcDES) showed, compared to a standard DES, an improved lipid extraction alongside a clear formation of several components that were identified as being primarily mono-acylated sugar esters. The following results show the enhancement of DESs made of sugar alcohols and choline chloride for the lipid extraction from *S. podzolica* biomass and subsequent sugar alcohol ester formation when supplementing acid and Novozym 435^®^ as biocatalyst in the system. The presented process can be carried under classical heating conditions, however, the microwave reactor provides accelerated media preparation alongside controlled heating and cooling conditions that end up in an eased handling of the viscous mixtures.

## 2. Results

The present work investigates and compares the preparation of several DESs (Table 1) obtained with a microwave dielectric heating versus a conventional convective thermal heating method. The extraction efficiency was investigated by the calculation of Fatty Acid Methyl Esters (FAMEs) and whole lipid per Cell Dry Weight (whole lipid/CDW), alongside the profiling of the lipids. Finally, structures of produced mono-acylated sugar alcohol esters were assessed using spectrometric and spectroscopic methods.

### 2.1. Comparative Study on the Production Time of Common and Sugar-Alcohol Based DESs

Figure 2 shows that formation of a DES with microwave heating was in all cases much faster than the thermal convective heating. Thus, the formation of a DES in the microwave can save considerable time, at least in such a scale. Considering the extremely short preparation times in the microwave, that were on average 16 times faster than in the incubator, it is reasonable to think that the energy input could be lower. Furthermore, choline chloride-based DESs were prepared on average 38% faster than betaine containing ones, regardless of the method.

Sugar-based DESs (1, 2, and 8) displayed overall longer preparation times, with either thermal or microwave heating. Such DESs are known to be difficult to prepare with conventional thermal heating and had to be produced with an adapted microwave method, since rapid heating was not possible. Adjusting the method automatically resulted in longer times to form such sensible DESs. Xylit DES and Sorbit DES, made with sugar alcohol, show drastically reduced formation times compared to the thermal heating method. Interestingly, sugar alcohol-based DESs were prepared faster than the non-reduced sugars, and observation tended to show that they were less susceptible to degradation.

Xylit DES and Sorbit DES were used to mediate the one-pot process, the sugar alcohols composing them also took part in the production of sugar alcohol esters. Next, we detail how the reaction’s crude was characterized and profiled post-reaction.

### 2.2. Post-Reaction Thin Layer Chromatography (TLC) of Glycolipids- and Non Glycolipids-Containing Mixtures

The TLC of Sorbit DES was presented in Figure 3. Similar profiles were obtained regardless of the DES. With a standard DES, with or without lipase, only a single spot could be identified (Rf = 0.08), corresponding to the sugar alcohol. In contrast, two spots with (Rf = 0.08 and 0.34) were found in an acidified DES without lipase. In the acidified DES with lipase, the same profile was obtained, but with higher intensity of the stain corresponding to the glycolipid (Rf = 0.34). A control reaction with an acidified DES with lipase, and fatty acid methyl esters (FAMEs) from single cell oil extraction, was performed for confirmation.

Chromatography purification performed thereafter allowed us to isolate each fraction containing either lipids or glycolipids.

### 2.3. Comparison of Extracted Whole Cell Lipid and Esterified Fatty Acids

To be quantified, the extracted fats have been esterified into methyl esters (FAMEs), and then, injected in a gas chromatography apparatus. The mean value of lipid content for all conditions carried out on the biomass in Sorbit DES or Xylit DES was calculated.

The amounts of FAMEs, whole lipids, and glycolipids per reaction were determined and are shown in Table 2. The highest amount of FAMEs and whole lipids was obtained by microwave DES preparation with acidified DES, followed by the similar condition with lipase addition. In addition, the sugar alcohol in the DES did not significantly affect the lipid, and the glycolipid, yields. In comparison, lipid extraction according to Folch (FE) and standard DES, with or without lipase, were far less efficient in this regard.

The FAMEs and whole lipids obtained per CDW are shown in Figure 4. A comparison of the different lipid extraction methods showed that direct transesterification of the oleaginous biomass to fatty acid methyl esters gave the highest FAMEs yield per CDW. Extraction according to Folch was less efficient in regard of the FAMEs and whole lipids contents. In comparison to DT, acidified Xylit DES extracted up to an honorable 70% FAMEs from robust yeast cells of *S. podzolica*. Intrinsically, our DESs were not the best extraction method reported however, the subsequent interesterification of those lipids for sugar alcohol ester production made the use of these unconventional media relevant.

Quantification of the various extracted fatty acids allowed then profiling of the lipid fraction.

### 2.4. Profiling of Extracted and Esterified Lipids

Lipid profiles were determined for each condition to assess if the fatty acid composition is influenced by the extraction method (Figure 5). When comparing the various fatty acid extraction methods, no significant differences in fatty acid profile were observed. In percentage terms, oleic acid (C18:1) with an average of 24.2% is the most common of all extraction methods, followed by palmitic acid (C16:0) with 9.0%, linoleic acid (C18:2) with 4.1%, and stearic acid (C18:0) with 2.8%.

Nonetheless, profiling helped rationalizing and elucidating the structure of the formed sugar alcohol esters. Indeed, the most occurring lipids were more likely to react and form the targeted compounds.

### 2.5. Structural Elucidation Using Spectroscopic and Spectrometric Methods

To analyze the structure of synthesized glycolipids, ^1^H-Nuclear Magnetic Resonance (NMR) and Mass Spectrometry (MS) were performed on the isolated glycolipid fractions. Chemical ^1^H-NMR shifts of sugar moiety (~4.11–3.51 ppm) and fatty acid chains (~2.25–0.80 ppm) were identified. MS performed additionally allowed clear structural identification of the mono-acylated sugar alcohols. The chemical shifts were as follows:

Glycolipid fraction with Xylit DES: ^1^H NMR (400 MHz, CD_2_Cl_2_) δ 4.17 (dd, J = 4.2, 2.0 Hz, 2H), 4.14 (dd, J = 3.8, 1.7 Hz, 3H), 4.15–4.08 (m, 3H), 4.12–4.06 (m, 2H), 4.02–3.92 (m, 2H), 3.90 (dd, J = 13.9, 3.5 Hz, 2H), 3.64 (dd, J = 9.6, 1.9 Hz, 1H), 2.24 (dq, J = 14.3 Hz, 4H), 1.97–1.89 (m, 2H), 1.26–1.19 (m, 38H), 0.80 (m, 9H). (Appendix A).

Glycolipid fraction with Sorbit DES: ^1^H NMR (400 MHz, CD_2_Cl_2_) δ 4.05 (dd, J = 12.4, 5.4 Hz, 1H), 3.75 (d, J = 6.6 Hz, 1H), 3.63–3.39 (m, 4H), 2.25 (t, J = 7.5, 2H), 1.95 (dq, J = 12.4, 6.2 Hz, 4H), 1.26–1.16 (m, 27H), 0.80 (m, 6H). (Appendix A).

MS was performed with ElectroSpray Ionization (ESI) (Appendix A). The results of the spectrometric analysis for the synthesis in the Xylit DES and Sorbit DES are shown in Appendix B
Table A1. Fourteen glycolipid fragments have been detected for the sorbitol-based DES. Thereof, 4 sorbitol palmitate, 4 sorbitol linoleate, 3 sorbitol stearate, and 3 sorbitol oleate. Twelve glycolipid fragments for the xylitol-based DES were detected. Thereof, 2 xylitol palmitate, 4 xylitol linoleate, 2 xylitol stearate, and 4 xylitol oleate.

## 3. Discussion

### 3.1. Microwave and DES Technologies: An Optimal Match?

In the present work, comparing the times required to produce common DESs with microwave and thermal heating, the efficiency of using a microwave to form these liquid mixtures was demonstrated. In theory, our approach could also significantly reduce processing time, energy costs, and equipment size compared to conventional convective or conductive thermal heating methods, but deeper investigations should be carried out to verify these claims [15,16,17]. Microwave assisted chemical processing requires the use of strong microwave absorbing solvents. Since the ability of some solvents, such as water, to interact in the electromagnetic field decreases sharply as temperatures rise, solvents with high thermal stability and low vapor pressure are inherently optimal [6]. DESs meet those requirements due to their polar nature and their outstanding number of hydrogen bonds. Indeed, as described by Nie et al., they are an ideal match for ionic conduction and dipole rotation occurring during heating. As a result, the energy emitted by the magnetron of the microwave can be absorbed optimally by the polar and ion-containing DESs [17,18]. Thus, we conjecture that a combined effect of both greater dipole rotation and ionic conduction allowed the ultra-fast preparation of the DESs presented in our study. Within the specific framework presented in this work, one-pot microwave treatment of oleaginous biomass could then represent a more economical substitute to pre-existing, notably mechanical, preparation procedures. Finally, and according to the literature, we did not expect thermal effects to have a significant role in the acceleration of DESs preparation or in the process itself [19,20]. However, in our case, further enzymatic kinetic studies comparing, for example, the influence of classical heating and microwaves, would be needed to fully prove this last claim as the matter remains controversial and unclear [21,22].

Microwave power input is however mainly limited by thermal decomposition of the DESs components [6]. Indeed, in high temperature treatments, sugar degradation may occur through a complex series of reactions known as the Maillard reaction or caramelization. During the latter, sugar molecules are broken down, and the reaction intermediates and reactants are then polymerized [23,24]. Thus, our method used 60 °C for 72 h, mostly in opposition to the literature that reports shorter reaction times but with temperatures reaching up to 150 °C.

In general, sugar based DESs exhibit high viscosity, density, and surface tension at room temperature. Thus, it is recommended to heat these DESs for processing, to ease their handling, since viscosity decreases when temperature increases [25]. Similarly, DESs containing betaine, instead of choline chloride, also tended to show higher times and difficulty to be formed. The addition of water to promote their formation was often required, as reported for betaine mixtures with glucose, urea, and small polyol molecules [26]. Microwave heating represents therefore a viable alternative to thermal heating for an efficient and scaled-up preparation for such mixtures. Sugars, such as glucose, sucrose, and arabinose could have been also tested for the glycolipids production, but the sugar alcohols were preferred because they were less represented in the literature. Moreover, we demonstrated that in combination with choline chloride, they were among the simplest and fastest DESs to prepare using microwave irradiation. Indeed, the presence of the hydroxyl function instead of an aldehyde decreases their reactivity for polymerization. In addition, Xu et al. could not detect any linearity between extraction yield and polarity for the extraction of flavonoids in a DES based on sugars. However, linearity could be found with DES made of sugar alcohols [27]. Reactivity of their primary alcohols was, because of the steric hindering, similar, thus hardly influenced the results. The use of DES media for extraction is then particularly worthwhile when at least one of its component undergoes conversion, thus creating a “2-in-1” system [28].

### 3.2. Unconventional Media for Lipid Extraction and Subsequent Production of Glycolipids

In this work, lipid extraction was achieved by treating the biomass with a combination of microwave, DES, and enzyme technologies that were individually reported as possible cell disruption methods [12]. Our designed one-pot extraction of lipids and production of mono-acylated sugar alcohol esters drastically reduced steps number. However, pretreatment of biomass by freeze-drying was still necessary, because moisture considerably reduced the yield of transesterification [29,30]. Indeed, moisture content of oleaginous microorganisms was over 80% [31]. In this work, highest lipid content could be measured with use of an acidified DES, followed by an acidified DES with addition of lipase. Since glycolipids were predominantly formed in an acidified DES with lipase, slightly less lipids have been measured as FAMEs. According to Gorte et al., it has been shown that by using acid instead of base, significantly more fatty acids could be obtained in the direct transesterification of the biomass. Yeast cell wall structure usually consists of two layers, including an inner layer with polysaccharides and an outer layer with glycoproteins covalently bound to the inner layer [32]. Thus, *S. podzolica* cells were very robust and needed harsh conditions for lipid extraction. That is why, in previous reports, mechanical methods such as high-pressure homogenizer and ball milling could recover higher yields of lipid per CDW and appeared more efficient that physical digestion with acidified DES.

Khoomrung et al. demonstrated the potential of a microwave-assisted technique for the rapid lipid extraction from yeast cells [33]. In addition, it has been shown that microwave pretreatment was one the most efficient methods among sonication, bead milling, and osmotic shock to extract lipids from oleaginous biomasses such as algae [34]. Carboxylic acid-based DESs seemed more suitable for the lipid extraction of oleaginous biomass, in comparison to sugar or polyol-based DESs [35]. This could be correlated to our results, which highlights that the sugar alcohol-based DES extracted significantly more lipids when acidified.

Lipase did not significantly influence the lipid extraction efficiency, since almost identical amounts of lipid contents were obtained in a standard DES, with or without lipase. However, lipase seemed to promote the glycolipid synthesis (Figure 3). An interesting aspect was the positive influence of HCl on the enzyme activity. It is known that Novozym 435^®^ is stable over a relatively wide pH range, especially in the alkaline one. Here, we demonstrated that glycolipids could be formed in an acidified DES with lipase. Thus, we speculated that inactivation of the lipase was limited in the presence of 0.1 M of HCl in DES, since it was shown that Novozym 435^®^ retained activity up to 0.3 M of acetic acid [36]. We conjecture that due to the high viscosity and low water activity/content of DESs [37], lipase activity was predominant as only a limited quantity of media could accumulate inside the porous acrylic carrier (Lewatit 1600) of Novozym 435^®^. Thus, the inactivation of the biocatalyst was rather minimal [38]. The conversion could partly occur due to acid or auto-catalysis, but based on our results, it is considered negligible [39,40,41]. ILs showed great efficiency toward lipid extraction as well, but imidazolium-based ones inhibited lipase activity [42,43]. Therefore, DESs represent a greener and simpler alternative in which biocatalysis can be performed [44,45]. Their association with microwave irradiation appeared interesting to form an effective duo using relatively mild conditions with overall low energy consumption [46].

During the flash chromatography purification, the glycolipid peak was only observed in the acidified DES with lipase, most likely because its concentration was too low to be detected by the light scattering detector (Appendix A). Glycolipids were therefore obtained in quantities that allow flash chromatography purification when both acidification and the enzyme were combined.

Based on our “2-in-1” deep eutectic system [28], with the microwave-assisted procedure, glycolipids can be tailor-made based on any sugar- (or polyol)-based DES with any lipid-containing biomass [47]. The current work using *S. podzolica* biomass made it possible to obtain long-chain glycolipids with no prior modification of the lipids (e.g., ethylation, methylation, or vinylation) [48]. These sugar esters find application in various fields, such as in the cosmetics industry (e.g., rhamnolipids, sophorolipids, and mannosylerythritol lipids) and in the pharmaceutical industry (e.g., mannosylerythritol lipids) [49]. Insects such as *Hermetia illucens* also represent an interesting source of oleaginous biomass as they are more susceptible to provide short-chain lipids, notably lauric acid, and myristic acid [50,51]. Thus, by using such a biomass, short-chained glycolipids could be formed. The production of laurate sugar esters raises interest, especially in the pharmaceutical industry where they are notably used to form drug delivery systems [52,53]. The properties of the glycolipids can be influenced not only by chain length but also by the ramifications and degree of unsaturation [54,55]. Furthermore, use of oleaginous biomass allows esterification of branched and unbranched fatty acids, that have different chain lengths and contain one or several alkene groups that represent gateways for further chemoenzymatic reactions [56].

## 4. Materials and Methods

### 4.1. Materials

Lipase formulation Novozym 435^®^ was given by Novozymes (Denmark). Lipid-containing yeast biomass was produced by fermentation as described below. All other chemicals were purchased either from Car Roth GmbH & Co. KG (Karlsruhe, Germany) or Sigma Aldrich Chemie GmbH (Taufkirchen, Germany) if not stated otherwise.

### 4.2. Microorganisms

The oleaginous basidiomycete used in this study was screened and deposited at the DSMZ culture collection (Deutsche Sammlung von Mikroorganismen und Zellkulturen, Brunswick, Germany) as *Cryptococcus podzolicus* DSM 27,192 by Schulze et al. [57]. After genome sequencing and annotation, the latter was phylogenetically reclassified to *Saitozyma podzolica* DSM 27,192 by Aliyu et al. [58].

### 4.3. DES Preparation with Microwave Dielectric Heating and Conventional Convective Heating

Formation of DESs was validated as reported by Dai et al. [59] and Hayyan et al. [25]. In a G30 Anton-Paar microwave vessel, DESs compounds were introduced according to the quantities in Table 1. The resulting mixture was manually homogenized with a spatula, then placed into the microwave synthesizer (Monowave 400, Anton-Paar, Ostfildern, Germany) where it was heated to 95 °C with slow heating mode for DESs made of actual sugars and rapid heating mode for the others (850 W, stirrer speed 600 rpm) with IR thermometer.

In a 50 mL Schott flask, DESs components were introduced according to Table 1. The resulting mixture was manually homogenized with a spatula, then placed into a Thermotron incubator (Infors HT, Bottmingen, Swittzerland) where it was heated to 95 °C at 300 rpm orbital shaking.

When clear homogenous phases were obtained, shaking and heating were stopped as the time was either given by the microwave software or simply noted from incubator’s timer.

### 4.4. Production of Single Cell Oil in Bioreactors

*S. podzolica* was cultivated in a 2.5 L Minifors bioreactor (Infors HT, Bottmingen, Switzerland) as described by Schulze et al. [57].

### 4.5. Microwave Processing of the Oleaginous Biomass

In an Anton-Paar G30 microwave vessel, were simultaneously introduced 10 mL of 5% 2M HCl (0.1 M) or 5% deionized water sugar alcohol-based DES with or without 200 mg of Novozym 435^®^ (Novozymes, Bagsværd, Denmark) and 400 mg of lyophilized yeast powder. The reaction was then heated at 60 °C as fast as possible for 72 h with the Anton-Paar Ruby Thermometer (Anton-Paar, Ostfildern, Germany) (850 W, stirrer speed 600 rpm until temperature was reached, then 300 rpm). As a control, fatty acid methyl esters (FAMEs) were used as lipid source for the glycolipid production as described by Siebenhaller et al. [10].

### 4.6. DownStream Processing (DSP) and Flash Chromatorgaphy Purification of the Reaction’s Crude

Once the reaction finished, the resulting mixture was dissolved in 10 mL of deionized water. The mixture was filtered with a Buchner funnel and the solid was washed with 3 × 5 mL water and with 3 × 5 mL of ethyl acetate. Then, 10 mL of brine was added, the aqueous phase was extracted with 6 × 25 mL of ethyl acetate in a separatory funnel, the organic phases were then gathered and dried over MgSO_4_ for 15 min. The solid chemical dryer was filtered-off with filter paper and the solvent was evaporated with a rotary evaporator to obtain a brownish paste.

The crude paste obtained was re-dissolved in chloroform to be adsorbed over 4 g of silica (Kieselgel 60) for flash chromatography purification using the solid loading method. To purify this crude, a Reveleris PREP purification system equipped with a 4 g Flash Pure spherical column was used. Elution solvents were chloroform and methanol with a gradient such as: 2nd solvent percentage started at 0% for 1.5 min, 7% for 9.5 min, 15% for 3 min, and finally 100% for 3 min. The first fraction was fatty acids collected at 3 min. The second fraction was glycolipids collected at 4.5 min. The collected fractions were gathered and dry evaporated on a rotary evaporator for further analysis.

The total lipids per cell dry weight (CDW) were then calculated using the following formula:(1)$$\%\mathrm{Lipids}\;\mathrm{per}\;\mathrm{Weight}\;\left(\mathrm{CDW}\right)\left[\%\right]=\frac{\mathrm{Lipid}\;\mathrm{extracted}\;\left[\mathrm{mg}\right]}{\mathrm{Weight}\;\mathrm{CDW}\;\left[\mathrm{mg}\right]}\times 100\%.$$

### 4.7. Folch Extraction and Direct Acidic Transesterification of the Biomass

To establish a base of comparison between methods for the extraction of lipids from oleaginous biomass, a Folch extraction was performed as follows. First, 400 mg of freeze-dried yeast biomass was homogenized by 45 sec vortex with 8 mL of chloroform/methanol (2/1) in a 15 mL Falcon tube. After dispersion, the whole mixture was agitated during 20 min in an orbital shaker at room temperature. The homogenate was filtrated with a funnel provided with filter paper. After centrifugation and siphoning of the upper phase, the lower chloroform phase containing lipids was dry evaporated under vacuum in a rotary evaporator to be analyzed and profiled using GC analysis.

As a reference for the lipid content comparison, direct acidic transesterification of the biomass was done, as described by Gorte et al. [11].

### 4.8. Acidic Transesterification to Fatty Acid Methyl Esters (FAMEs) of the Lipid Fraction

The dried lipid fraction purified by the flash chromatography was dissolved in 1.5 mL of hexane. The solution was transferred to a glass tube and 0.5 mL of the internal standard (2 mg/mL heptadecanoic acid) and 2.0 mL of 15% H_2_SO_4_ in methanol were added. In addition, a blank was prepared in a glass tube consisting of 1.5 mL hexane, 0.5 mL internal standard, and 2.0 mL 15% s H_2_SO_4_ in methanol. The glass tubes were then incubated for 2 h at 100 °C and 1000 rpm in a thermal shaker. In between, the samples were homogenized by hand at intervals of about 30 min. After incubation, the glass tubes were cooled on ice for at least 5 min. Then, 1 mL of MilliQ water was added to the glass tubes and centrifuged at 2500 rpm for 5 min. Finally, 1 mL from the upper phase was transferred to a GC vial.

### 4.9. GC Analysis of Fatty Acid Methyl Esters (FAMEs)

The quantitative and qualitative analyses of the transesterified FAMEs were performed gas-chromatographically using the 6890 N Network GC-System (Agilent Technologies Deutschland GmbH, Waldbronn, Germany). The device was coupled with a DB-Wax column (122–7032; l: 30 m d: 0.25 mm, Agilent Technologies Deutschland GmbH, Böblingen, Germany) and the detection was performed with an FID under 1.083 bar working pressure. Then, 1 mL of sample was injected at the initial temperature of 40 °C. The separation of the FAMEs was achieved by a temperature gradient from 40 to 250 °C with a rate of 8 °C/min and was kept for 10 min at 250 °C. To identify and quantify the FAMEs, the RM3 FAME Mix standard (07256-1AMP; Sigma Aldrich, Taufkirchen, Germany) was used. Chromatogram and integration of the main signals are given in Appendix A.

The yield of FAMEs and the FAMEs per CDW [%] were then calculated using the following formulae:(2)FAMEs [mg]=extracted whole lipids [mg]×FAMEs per whole lipid [%],
(3)$$\mathrm{FAMEs}\;\mathrm{per}\;\mathrm{CDW}\;\left[\%\right]=\frac{\mathrm{FAMEs}\;\left[\mathrm{mg}\right]}{\mathrm{CDW}\;\left[\mathrm{mg}\right]}\times 100\%.$$

### 4.10. Thin Layer Chromatography (TLC) Analysis of Reaction Mixtures

After the DSP of the resulting mixtures, 10 mg of each reaction´s crude was dissolved in 1 mL of chloroform/methanol (75/25) and further used for TLC analysis as follows. First, 10 μL of the previously extracted organic phase was spotted onto a silica plate (Alugram SIL G, 60 Å, Macherey-Nagel GmbH & Co. KG, Düren, Germany). The eluent consisted of chloroform: methanol: acetic acid (65:15:2 *v*/*v*) [60]. After elution, TLC plate was dived into anisaldehyde: sulfuric acid: acetic acid (0.5:1:100 *v*/*v*) dying solution and subsequently revealed with a heat gun.

### 4.11. Spectroscopic and Spectrometric Methods for Structural Eluciation of Glycolipids

The nuclear magnetic resonance (NMR) spectroscopy spectra were measured in dichloromethane-*d_2_* (CD_2_Cl_2_ purchased from Eurisotop, Saarbrücken, Germany) on a Avance 400 NMR instrument (Bruker Biospin GmbH, Rheinstetten, Germany), with 400 MHz for ^1^H. The chemical shifts are expressed in δ (ppm) versus tetramethylsilane (TMS) = 0. The mass spectrometry (MS) for mass identifications was performed with electrospray ionization (ESI) on a quadrupole Q Exactive Plus (ThermoFisher Scientific GmbH, Kandel, Germany) and recorded in positive mode. Raw data from MS and NMR was treated using MestReNova Suite 2020 [version 14.2.0].

### 4.12. Statistical Analysis

OriginPro software 9.7 [version 2020] (OriginLab Corporation, Northampton, MA, USA) was used for raw data treatment and statistical analysis. One-way ANOVA followed by post hoc test Tukey were performed using *p*-value < 0.05.

## 5. Conclusions

In this work, we demonstrated a proof of concept of the lipid extraction from oleaginous biomass and the subsequent glycolipids formation using lipase-catalyzed reaction with microwave-assisted processing in acidified DES. The medium made of choline chloride and sugar alcohols was prepared, among other more common DESs, in a matter of seconds using microwave technology, highlighting a technological match. In such “2-in-1” system, any combination of sugars and lipids is virtually possible to prepare tailor-made glycolipids. Indeed, other oleaginous biomasses for the synthesis of glycolipids could be considered. With the help of the developed robust reaction system combined with a proper identification strategy, compounds were isolated and characterized. In addition, screening for potential bioactivity of the DES containing mixtures or isolated compounds would be an interesting follow-up.

However, only a limited amount of glycolipid was obtained making optimization required. It is therefore important to investigate higher temperatures, different lipase formulations, and develop kinetic determination methods to allow an optimized scale-up of our process. In addition, chloroform and methanol were used as solvents for the purification of the glycolipids by flash chromatography. The replacement of the latter by halogen-free, more environmentally friendly solvents, such as hexane and ethyl acetate, or developing alternative methods to get rid of chromatographic purification should be considered.

The acidification of DESs to promote the lipase-catalyzed reaction must be understood in the future. Finding how much glycolipid can be produced by the specific activity of the lipase and how these conditions affect the overall activity and stability of the lipase formulation, as well as the recyclability of the biocatalyst, should all be investigated.

## Figures and Tables

**Figure 1 molecules-26-00470-f001:**
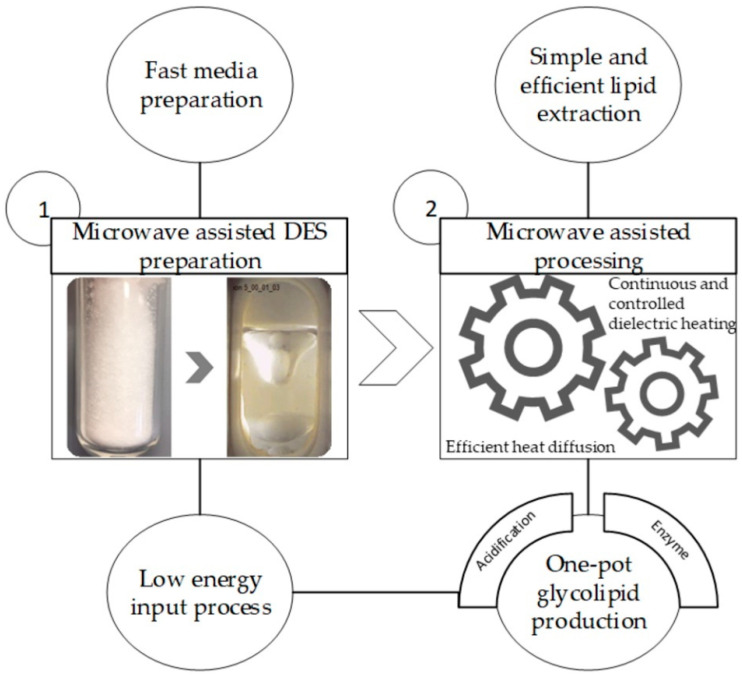
Flowchart of this study. Freeze-dried oleaginous biomass from *S. podzolica* was used in a one-pot microwave-assisted process that extracted fatty acids and subsequently produced a purifiable quantity of glycolipids when lipase and acidified Deep Eutectic System (DES) were jointly used.

**Figure 2 molecules-26-00470-f002:**
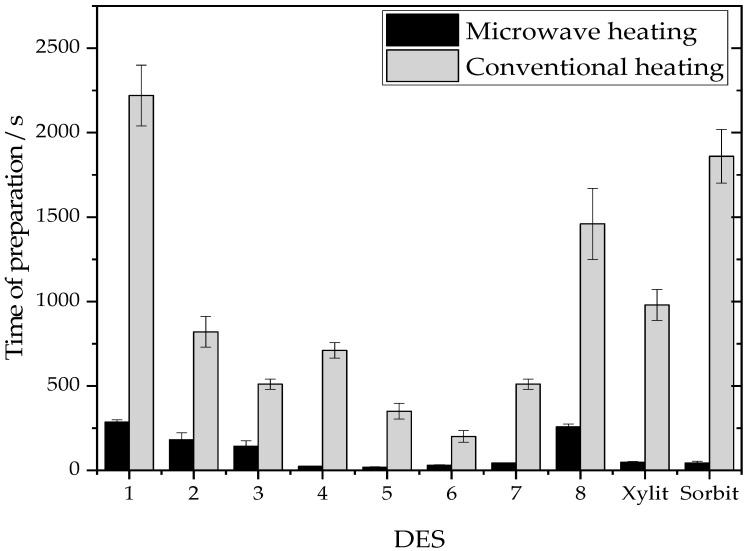
Preparation times of common and sugar-alcohol DESs with microwave heating and thermal heating, mean standard deviation indicates significant differences (*p* < 0.05).

**Figure 3 molecules-26-00470-f003:**
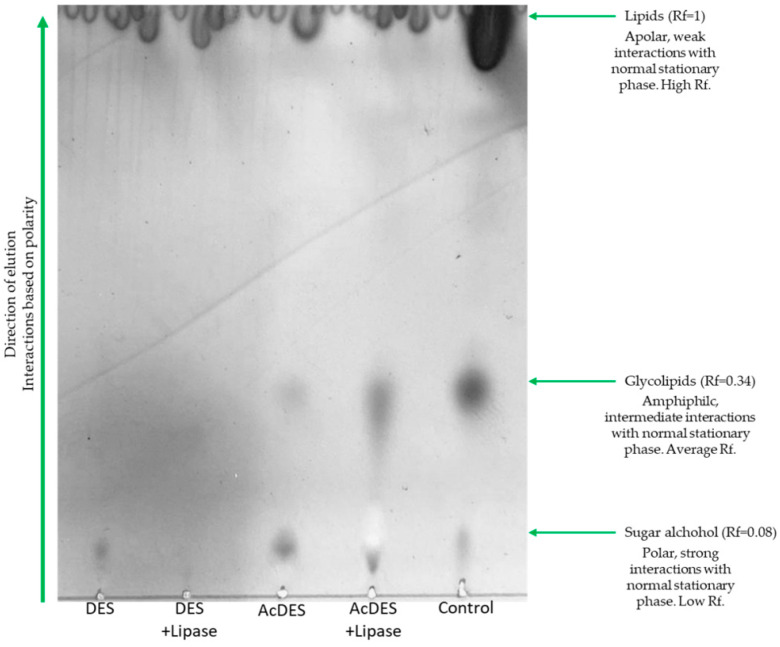
Thin layer chromatography of the reactions processed with microwave irradiation including a control using directly Fatty Acid Methyl Esters (FAMEs) from yeast biomass. AcDES: acidified DES; DES: standard DES.

**Figure 4 molecules-26-00470-f004:**
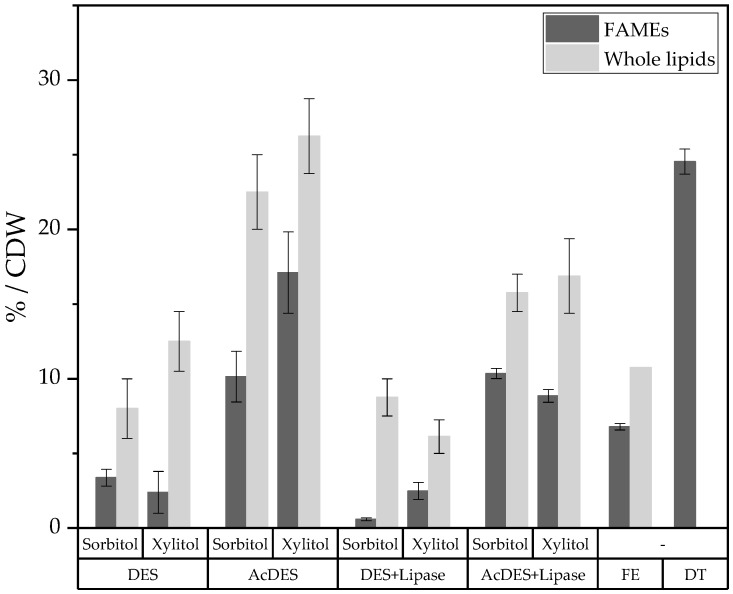
Fatty Acid Methyl Esters per Cell Dry Weight (FAMES/CDW) [%] for the microwave extraction of fatty acids under different reaction conditions, the Folch extraction process (FE), and the direct transesterification of the oily biomass to FAMEs (DT). Mean standard deviation indicates significant differences (*p* < 0.05).

**Figure 5 molecules-26-00470-f005:**
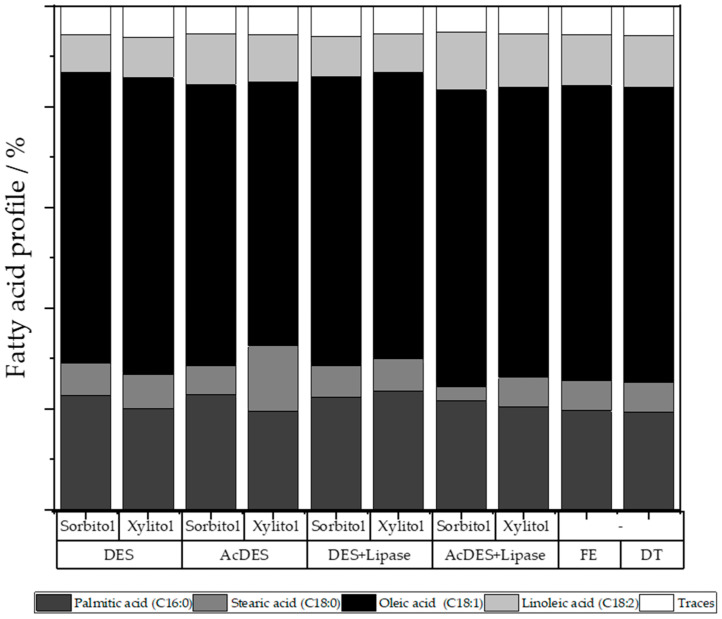
Distribution of fatty acids for the extraction carried out by microwave with different sugar alcohol-based DES conditions, the Folch extraction process (FE), and the direct transesterification (DT) on the oily biomass.

**Table 1 molecules-26-00470-t001:** Composition of common and sugar alcohol-based DESs studied.

DES Number/Name	HBD	mR [mol]	m_HBD_ [g]	HBA	mR [mol]	m_HBA_ [g]	mR [mol]	m_water_ [g]	wt % Water
1	Arabinose	1	6.45	ChCl	1	6.00	0.8	0.62	5
2	Glucose	1	5.00	ChCl	2	9.69	2.50	1.25	9
3	Glycerol	2	7.86	Betaine	1	5.00	1.00	0.77	6
4	Urea	2	4.30	ChCl	1	5.00	-	-	-
5	Glycerol	2	6.60	ChCl	1	5.00	-	-	-
6	1,2-Propanediol	1	2.54	ChCl	1	4.65	1.00	0.60	8
7	1,4-Butanediol	4	6.15	Betaine	1	2.00	1.00	0.31	4
8	Saccharose	1	5.00	ChCl	4	8.16	4.00	1.05	8
Xylit	Xylitol	1	5.45	ChCl	1	5.00	0.8	0.52	5
Sorbit	Sorbitol	1	6.53	ChCl	1	5.00	0.9	0.58	5

Note. Final volume of DES results in 10 ± 1 mL, HBA: hydrogen bond acceptor, HBD: hydrogen bond donor, mR: mole ratio.

**Table 2 molecules-26-00470-t002:** Amounts of extracted whole lipids, FAMEs, and glycolipids per 10 mL of reaction.

Condition	DES		AcDES		DES + Lipase		AcDES + Lipase		FE	DT
Sugar Alchohol	Sorbitol	Xylitol	Sorbitol	Xylitol	Sorbitol	Xylitol	Sorbitol	Xylitol	-	-
Extracted whole lipids ** (total amount (mg))	32 ± 1	50 ± 2	90 ± 1	105 ± 1.5	35 ± 1	25 ± 1	63 ± 1	68 ± 3.3	43 ± 1	-
Extracted FAMEs ** (total amount (mg))	14 ± 1	10 ± 2	45 ± 3	68 ± 3	2.4 ± 0.1	10 ± 1	40 ± 1	35 ± 1	27 ± 1	5 ± 1 ***
Glycolipid quantity (total amount (mg))	-	-	T *	T *	-	-	~15	~20	-	-

Note. * T stands for Traces meaning observation of glycolipid production could be made on TLC, but the amount was too low for chromatography purification. ** All values are given as the mean ± standard deviation of at least three independent experiments using each 400 mg of freeze-dried biomass that showed statistically significant differences, *p*-value was < 0.05. *** Direct transeterification was done with 20 mg of freeze-dried biomass and only produced FAMEs.

## Data Availability

The main data presented in this study are available in the Appendix A.

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
