# Peer review of "Microwave-Assisted One-Pot Lipid Extraction and Glycolipid Production from Oleaginous Yeast Saitozyma podzolica in Sugar Alcohol-Based Media"

_molecules, 2021, doi:10.3390/molecules26020470_

Round 1

Reviewer 1 Report

The authors peresnt a new method of glycolipid production  from oleaginous yeasr Saitozyma podzolica  which calls for one-pot extraction. This simple method results in acceleration on whole process. The method allows subsequent conversion of lipids into mono-acylated palmitate, oleate or stearate sugar alcohol esters. This is a new and quite interesting method.

Minor issues:

Figure 1 could be improved by using smaller font. 32

Author Response

We hereby thank the reviewer for his comment:

Figure 1 was changed using accordingly smaller font. (L93)

Reviewer 2 Report

This study investigated the extraction of lipids from oleaginous biomass and the subsequent formation of glycolipids using a lipase-catalyzed reaction exposed to microwaves in acidified DES. The conducted research is undoubtedly a topical direction of science. However, there are some issues need to be addressed.

 My major comment is connected with the structure of the manuscript. In my opinion, essential part of the information of Discussion section, especially, subsection 3.1, should be moved to the Introduction section. In present form, the Introduction section give few information on the problems to be solved.

Other comments.

  1. Section 4.2 describes the preparation of DES without a literature source. Can the authors of this work provide any experimental data confirming the formation of DES? Is it not possible that in a pure homogeneous solution, after the formation of which the authors stopped shaking and stirring, decomposition products of HBD and HBA were formed, especially under the influence of microwaves?
  2. It is interesting to know about the stability of extracted lipids. Is it possible for them to decompose under the action of a high-pressure homogenizer? This question is very important in this paper, since the extracted lipids are used in the further synthesis of glycolipids. I would like to see spectroscopic and spectrometric data for structural elucidation not only for glycolipids, but also for extracted lipids.
  3. In a GC chromatogram, the area under the peak is usually used to determine the concentration of the sample components by comparison to a determined calibration curve. The linearity of concentration of solutes established by analyzing calibration standard samples at different concentrations should be presented in Supplementary Material.
  4. There are grammatical errors in the paper. Some places are very difficult to understand. A thorough review of the entire text of paper is needed.

5. Quality of Figures should be improved.

Author Response

We hereby thank the reviewer for his comments:

Portions of the Discussion were moved accordingly to the Introduction (L52-53/56-58/66-70). This allowed in the meantime to expose the problematics and the solutions more clearly.

  1. References for DESs preparation method were added to the concerned subsection (L337). DESs used in this study have been studied extensively in the literature for various parameters (cytotoxicity, stability, conductivity…). We do not exclude that degradation might have happened however it is considered herein to be minor and plays therefore neglectable role over the presented process. Indeed, temperatures used in this work (95°C) are relatively far from the decomposition temperature of polyols/sugars (~ 200°C) and ammonium salts used (up to ~300°C). The highest concern for degradation being the sugar (addressed L234-240).
  2. It is a very valid remark from the reviewer. Given the spectroscopic and spectrometric data of the produced sugar alcohol esters, it is likely to think that the integrity of the fatty acids was preserved. We do not refute that degradation doesn´t occur to an extent but comparing fatty acid profiles (Figure 5) from direct acidic transesterification (DT) and the DES conditions acidified or not, we obtain similar results based on our GC analysis using the commercially available FAMEs-containing standard (RM3 FAME Mix standard, Sigma-Aldrich). Biomass pre-treatment using high pressure homogenizer from previous report (Gorte et al. 2020) were compared to ours since we use similar analysis method. Despite higher yield of lipid extracted per CDW the fatty acid profile remained the same. L185-187.
  3. The method used herein for quantification was previously established and reported in several papers dealing with lipid extraction from oleaginous biomass as well (Schulze et a 2014, Gorte et al. 2020…). It is an internal standard method using heptadecanoic acid (non-native, odd chain fatty acid) at 2 mg/mL of which 0.5 mL were added to each sample (as specified L386-388 of the Materials and Methods section). The RM3 Fame Mix standard is used then to identify each methylated fatty acid and its respective retention time. In Supplementary Material was added as an example the chromatogram of the experimental FAMEs mixture and its treatment via signal integration (Supplementary Figure S5)L404.
  4. Manuscript has been revised thoroughly to correct grammatical errors and paragraphs that were hard to understand.
  5. For an overall improvement of the manuscript, figures containing treated data were transferred to OriginPro (version 2020). The rest of the figures were also improved by changing colors, fonts and additional information.L93/124/139/169/181/425.

Reviewer 3 Report

From my point of view it is a very interesting, innovative and highly applicable work.

I only have a couple of comments to make to the authors.

- In figure 3, next to the image of the TLC I would place a drawing of it, to facilitate its understanding.

- In table 2, I do not think it is correct to use mg/reaction as the unit of concentration. They should be changed to "mg/equivalent sample quantity used" for example.

- In section 2.4 it would be necessary to include a standard chromatogram of the fatty acids obtained.

Author Response

We hereby thank the reviewer for his comments.

  • Figure 3 (TLC) was modified with an additional drawing that shows the direction of elution and a short text explaining migration of the stains based on polarity and interaction with stationary phase.L139
  • In Table 2 we agree that “mg/reaction” could be changed. We propose therefore “total amount (mg)”.L156
  • Chromatogram was furnished alongside the concentration and area of each analytes used for the GC quantification in Supplementary Material. Supplementary Figure S5. L404.

Reviewer 4 Report

In my opinion, manuscript ID molecules-1060861entitled “Microwave-assisted one-pot lipid extraction and glycolipid production from oleaginous yeast Saitozyma Podzolica in sugar alcohol-based media” is correctly written, concerns an interesting and practical issue and fits into the profile of Molecules . I believe that the Authors took great care in designing the experiments, conducting them and presenting the results in a reliable way.

The Introduction provides a good background and inspiration for the research work undertaken. The methodology section is clearly written. The Authors used appropriate statistical methods to demonstrate the significance of differences between the variables. They also used up-to-date literature for the Introduction and Discussion.

My comments below:

  1. Abstract background is too developed. Authors should limit the information from the literature review and to a greater extent take into account and present the obtained results. The abstract should contain also the most important conclusions.
  2. A graphical abstract is required in order to better understand the authors' intentions and the course of action during the conducted research.
  3. In the Introduction section, the innovative character and originality of the proposed solution should be presented in a more unambiguous way. Is the use of electromagnetic microwave radiation the only new element? Can similar effects be achieved with other conventional heating methods? Did the authors expect to get any athermal effects of microwave radiation? Is this extraction method economically justified, since the microwave efficiency is low? This should be described in detail in the introduction.
  4. Apart from the clearly defined aim of the experimental work, the authors should define the research hypotheses that they tried to verify during the conducted research.
  5. The authors write that it is a "low energy imput method". Please, compare the energy inputs and the final effects obtained with other methods: mechanical preparation, ultrasonication, high – pressure homogenization, hydrotermal depolimerisation, conventional heating, rtc.
  6. Please explain why microwave heating turned out to be faster and more effective (Fig 2). This should also be related to the required energy inputs and related to the economic aspect of the process.
  7. Fig 4 is very illegible. Font should be increased.
  8. Discussion section. Line 183-184: "It can also significantly reduce processing time, energy costs, and equipment size compared to conventional convective or conductive thermal heating methods [12-14]." This should be proved with reference to own research. The efficiency of converting primary energy to microwaves energy in the case of magnetrons is only about 50%. I do not entirely agree with this sentence. It has to be proven.
  9. In the methodology, the Experimental design subsection would be very useful. At the beginning, the concepts of research and the division of experiments into stages, series and research variants should be provided. Perhaps a diagram or table would be useful in this case.
  10. Were any optimization methods used and on what basis were the experiments planned (experimental design methods?).
  11. The significance of the group differences between the analyzed variables, the normality of the distribution of the variables, all of this. One-way ANOVA followed by post hoc-test Tukey were performed using p-value <0.05 also using Excel software? This is very surprising. Please explain and / or supplement the methodology.

Author Response

We hereby thank the reviewer for his comments.

  1. Abstract has been shortened accordingly and focused on obtained results alongside the most important conclusions. L16-31
  2. A graphical abstract was originally furnished as the first submission was done. Editor has been contacted concerning this issue. The answer: “In the Susy system. the GA is not available to reviewers, but to the Academic Editor. And we will send the GA to the reviewer during the revised review stage.”
  3. In the Introduction purpose has been explained more clearly L48-91. In the discussion section points such as: microwave dielectric heating over conventional heating sources to carry out the process has been developed (L91-93/L232-233). Non-thermal effects expectation was also discussed (L230-232), and economical justification of the presented method has been nuanced (L215-218/L228-229). Indeed, we do not conjecture herein that using microwave heating is per se a low energy input method but that the whole process is, as we removed cell disruption step prior to lipid extraction. We agree that these assumptions should be confirmed through own experiment via potential follow up studies on energy consumption, however we deem the hypothesis to be plausible. This has been made clearer in the text.
  4. Research hypotheses that led the presented results and methods have been stated accordingly in the introduction. L80-91
  5. Comparison of energy inputs between different treatment method was made based on information available literature. It is indeed important to note that our process, despite less effective regarding extraction yields, does not require prior treatment of the biomass for cell disruption like most reported methods. L66-70
  6. Conjectures on the role of microwaves to accelerate DESs formation have been formed and correlated to economic aspects.L225-227
  7. Figure 4 as well as other figures presenting graphs were re-worked using OriginPro 9.7 (version 2020). L93/124/139/169/181/425.
  8. The reviewer makes a very justified comment however we only make conjectures based on parallel research concerning economic aspects. “can” will be then changed to “could”. In the concerned sentence. Correction and nuance have been brought to the emitted hypothesis. L215-218
  9. We conceded on Comment #4 that presenting the hypotheses and expectations that lead the work provides a succinct overview of the different variants tested. Moreover, the graphical abstract will give more insight on the directions that the presented work took. L80-91
  10. We present here a proof of concept that open possibility for follow up studies on optimization and improved understanding of the process as we state in the conclusion. We thereby established a method that will be later potentially subjected to more advanced experimental designs allowing deeper understanding of the role held by each engaged parameter.
  11. Statistical analysis on Excel was done installing several plugins and following procedures available online. In order to remedy to the betterment of the overall appearance of the figures and for scientific relevance, all data treatment and identical statistical analysis were transferred to OriginPro 9.7(version 2020). L425-426

Round 2

Reviewer 2 Report

The authors have addressed most of the points raised after the first review and I am happy they could tell a more convincing story from their important work. Some questions remain, but I think as it stands this paper is acceptable and should lead to future developments. And one more small note: please, eliminate the punctuation errors in the manuscript.

Reviewer 4 Report

Thank you to the authors for responding to my comments. In my opinion, the manuscript can be published in present form.